# Comparative Study on the Origin and Characteristics of Chinese (Manas) and Russian (East Sayan) Green Nephrites

**Jiaxin Wang and Guanghai Shi ***

School of Gemology, China University of Geosciences Beijing, Beijing 100083, China; 2018810028@cugb.edu.cn
* Correspondence: shigh@cugb.edu.cn

**Abstract:** Green nephrites are widely pursued for their mild texture and vivid color. In recent years, many Russian green nephrites appeared in China (the world's largest nephrite market) and competed with the Chinese Manas green nephrites, which are traditionally highly valued. In this study, we compared the appearance, mineralogy and geochemical features (with EPMA and LA-ICP-MS) of Chinese (Manas) and Russian (East Sayan) green nephrites to objectively characterize and distinguish between these two nephrites. Chinese (Manas) and Russian (East Sayan) green nephrites are mined from serpentinized ultramafic units in the northern Tian Shan and East Sayan orogen, respectively. In terms of appearance, the Manas green nephrites are slightly bluish or grayish, whilst their East Sayan counterparts are brighter (duck-egg cyan). The Manas nephrites commonly have a caramel color, crumple structure, characteristic white globules and sinuous veins, green stains and yellow–green veins, together with a local fibrous structure. The East Sayan green nephrites are more transparent, with a gentler fine texture, uniform color, many black spots and a few green spots. Some green nephrites from the Arahushun mine of East Sayan have an ice-like appearance. Microscopic petrography and EPMA analysis indicate that both the Manas and East Sayan green nephrites comprise mainly tremolite with minor actinolite. Minor minerals in the Manas samples include chromite, chlorite-group minerals, and uvarovite; whilst those in the East Sayan samples include actinolite, chromite, chlorite-group minerals, and bornite. Bornite is not found in any other sources of green nephrite, and thus is characteristic of Russian (East Sayan) green nephrites. LA-ICP-MS trace element data in their amphiboles and Single-Factor Analysis of Variance (ANOVA) results suggest that the differences in Cr, Zn, Y, Ba, and Sr contents and values of $\delta$Eu, Eu/Sm, $(La/Yb)_N$, $(La/Sm)_N$, $(Gd/Yb)_N$, $\sum$HREE, $\sum$LREE/$\sum$HREE between the two nephrites are present, and can be used as their origin trace.

**Keywords:** green nephrite; Manas; East Sayan; origin identification; tremolite; trace element geochemistry

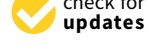

## 1. Introduction

A green nephrite is an assemblage of greenish amphibole minerals [1], and is famous for its hardness, mild texture, and beautiful color. It is often used for bracelets, pendants, and objets d'art. It has always been an important type of jade (nephrite and jadeite) in East and Southeast Asia. For instance, the green nephrites from Manas (NW China) were mined at an industry scale during the Ming Dynasty (1368 to 1644 A.D.) as royalties, and are commonly found in ancient royal collections and archaeological mining sites [2]. However, the geologic/geographic origin of green nephrites in many ancient/modern artifacts is hard to determine, which hampers research in archaeology, history, and culture. New sources of green nephrites emerged in recent decades from countries such as Russia, Canada, and New Zealand [3], among which the Russian ones were praised for their large size, good quality and color, and fewer cracks and black spots. Different green nephrites have a different geologic origin, quality, and culture. Their market value is also different; hence, identifying the origins of a nephrite is also highly important for consumers and collectors [4].

The Manas County (Changji Prefecture) in Xinjiang is located in the northern Tian Shan Mountains, south of the Zhunger Basin, at an altitude of about 3000–3500 m, and along the Tian Shan snow line. The main exposed strata at the mine are Devonian and Carboniferous strata, and the wallrocks mainly comprise mafic volcanics and tuff. The intrusive rocks in the mine are largely ultramafic, including plagioclase peridotite and plagioclase pyroxenite [5–8]. The majority of nephrite deposits in Russia occur in the folded belt of the southern Siberian Craton, which can be roughly divided into four nephrite producing areas: West Sayan, East Sayan, Djida and Vitim. Among them, green nephrites are mainly produced in East Sayan, southeast of Lake Baikal. At present, there are over 10 nephrite deposits being mined in the ophiolite belts of the East Sayan Mountains, among which, the most famous ones are No. 7 (Ospo), No. 10 (Gorlykgol), No. 11, No. 37, and Arahushun mines. Tang et al. [9–12] suggested that the East Sayan green nephrite deposits are products of ultramafic rock alteration, similar to those in Manas, New Zealand, Canada and Hualian (Taiwan). Zhao et al. [13] considered that the jade was formed by metasomatism, first by ultramafic rock serpentinization and then by contact metasomatism of the serpentinite with volcanic wallrock. The latter process obtained Si and Ca from the wallrocks, forming tremolite through diopside and tremolite alterations [13].

Zhang et al. [14–18] studied the gemological characteristics and basic structure of Russian (Gorlykgol) green nephrites, and suggested that the colors ranged from light to dark green, with minor yellowish or bluish tones. The Gorlykgol green nephrites commonly have dark spots and waxy–greasy luster, and range from translucent to opaque. These nephrites have a relative density of 2.93 to 2.98 (avg. 2.96), a hardness of 5–6, and a slightly lower refractive index (1.59–1.61, avg. 1.60) than the standard nephrite. They mainly have a fibroblastic structure, and are occasionally flaky. Zhao et al. [13] suggested that the main mineral composition of Ospa (No. 11) green nephrite (Russia) is tremolite and the black spots are chromite (via Electron probe microanalysis (EPMA) data). Yuan [19] identified chlorite around chromite in the Ospa green nephrites, and suggested that the cat's eye effect is caused by parallel needle-like tremolite. The chroma of green nephrite is mainly related to the Cr content.

Despite these studies [20–38], there are still many unanswered questions regarding the origin of these nephrites from China and Russia, including (1) whether Manas green nephrites comprise mainly actinolite or tremolite; (2) the mine that the nephrites samples from the market originate from; (3) indicators to distinguish the green nephrites from Manas and East Sayan. In this study, we attempted to tackle these questions through an integrated study on the nephrite morphology, mineralogy, and EPMA/LA-ICP-MS geochemical compositions.

## 2. Materials and Methods

### 2.1. Materials

In this study, green nephrite samples were collected from Manas (QKT, n = 14, QKT is the acronyms of name of the mine: Qiekuotai) and East Sayan (RU, n = 11, RU: Russia), including from No. 7 mine (7#, n = 4), No. 10 mine (10#, n = 2), No. 37 mine (37#, n = 2), Arahushun mine (AR, n = 2), and Arahushun pebble mine (AR-p, n = 1). Photos of these rough stone samples are shown in Figure 1. The samples were cut to cylinders (2.5 cm diameter, 1 cm thick), and the parts with discernibly different colors and structures or impurities were prepared separately in cylinders. A total of 65 and 25 cylinders were cut from the Manas and East Sayan samples, respectively. Representative sample photos are shown in Figure 2.

### 2.2. Methods

The color, texture, structure, impurities and other morphological features of the green nephrite samples were observed as hand specimens.

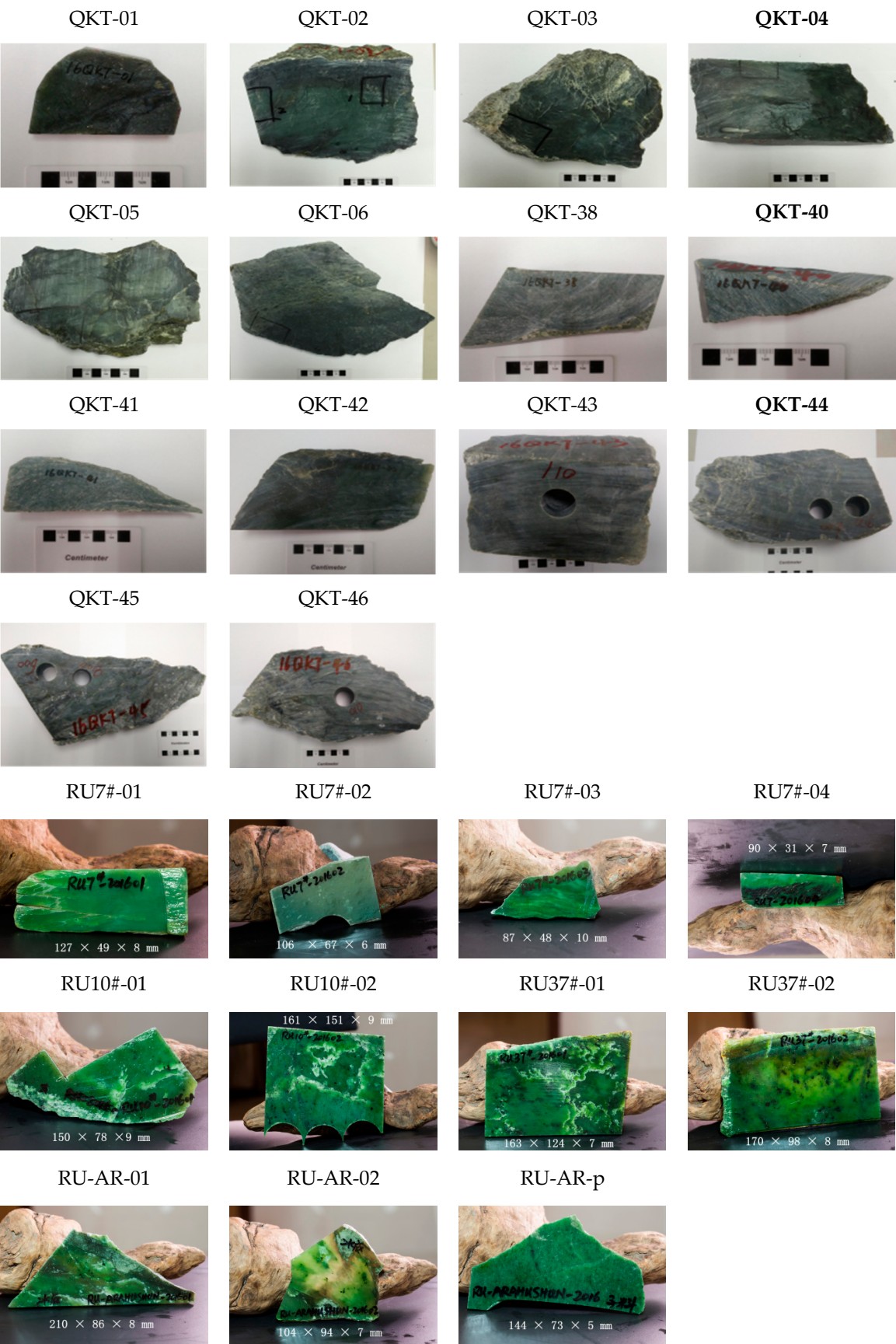

**Figure 1.** Photos of the Manas and East Sayan green nephrite samples.

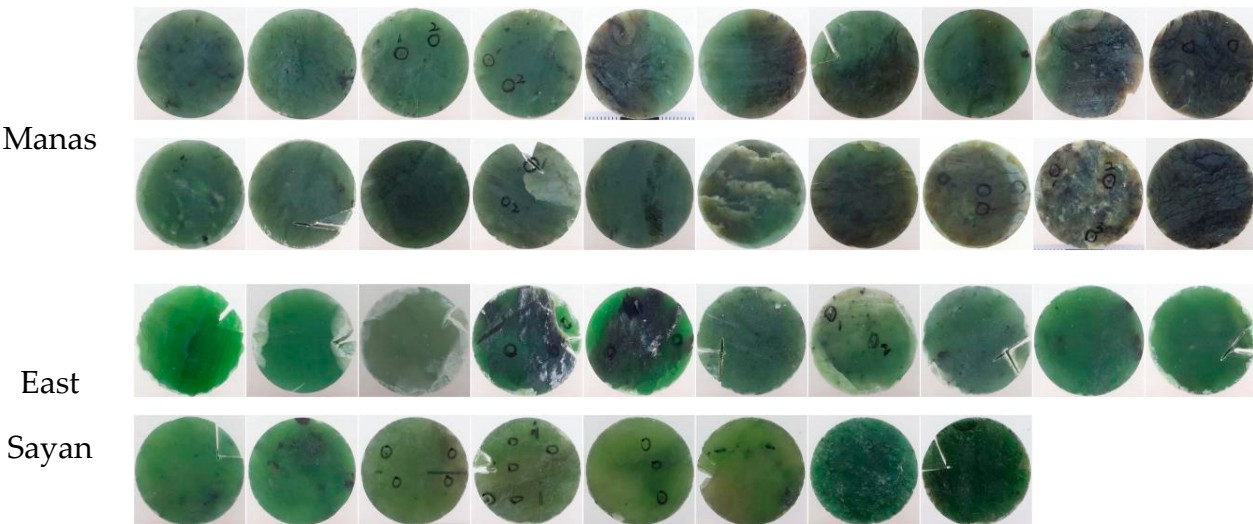

**Figure 2.** Color and textural features of the cylindrical mounts of the Manas and East Sayan green nephrites (2.5 cm diameter) from the same samples as in Figure 1 (The black circles in the figure is the orientation of EPMA test).

Electron probe microanalysis (EMPA) was conducted in the Electron Microprobe Laboratory of the China University of Geosciences (Beijing, China), using a Shimadzu EPMA-1600 Electron Probe Microanalyzer (Shimadzu, Kyoto, Japan). Analytical conditions included 15 kV acceleration voltage, $1 \times 10^{-8}$ A beam current, 1 μm beam spot size, and ZAF modification method. The standards used were Si, Al, Na (albite), Ti (rutile), Fe (almandine), Mn (rhodonite), Ca (calcite), and K (sanidine). The crystal chemical formula calculation method was as follows: 23 oxygen atoms were used as reference for EPMA data; minimum $Fe^{3+}$ was estimated using the 15eNK method.

LA-ICP-MS analysis was performed to measure the trace element compositions. The analysis was conducted at the Institute of Geology and Geophysics, Chinese Academy of Sciences (IGGCAS), Beijing, China, using a GeoLas laser ablation system coupled with an ELEMENT XR mass spectrometer. Analytical conditions included 11.1 J/cm$^3$ energy density, trace element standard NIST612, internal standard $SiO_2$, and 44 μm spot diameter. Other analytical parameters and procedures are described in reference [39].

The Analysis of Variance (ANOVA) is a statistical method for comparing the mean of variables in two or more independent groups. One-way ANOVA used only one variable to compare different groups, focusing on the inter- and intra-group variation. To test whether the trace element contents of the Manas and East Sayan samples have statistical differences, we used the one-way ANOVA approach. If the intergroup difference was greater than the intragroup difference, the former was considered to be significant, i.e., not caused by errors produced by different testers, and thus their classification and the comparisons between them were meaningful [38].

## 3. Results

### 3.1. Morphological Features

Munsell color theory is the most widely accepted color theory in the gemological field. According to this theory, colors are mainly influenced by hue, value and saturation (Figure 3). Hue represents the type of color, e.g., red, blue, green. Lightness refers to the brightness, e.g., bright, dark (higher lightness = brighter). Saturation refers to the concentration of a color (higher saturation = thicker). The color perceived by the human eye is the combined effect of these three factors.

The primary hue of Manas green nephrite is green, and the secondary hue is usually blue or gray, with a low lightness and medium saturation. The main hue of East Sayan green nephrite is green, with a bright lightness and thick saturation. Some samples exhibit

grey as the primary hue and green as the secondary hue, with a high lightness and low saturation, commonly known as duck-egg cyan (Figure 4).

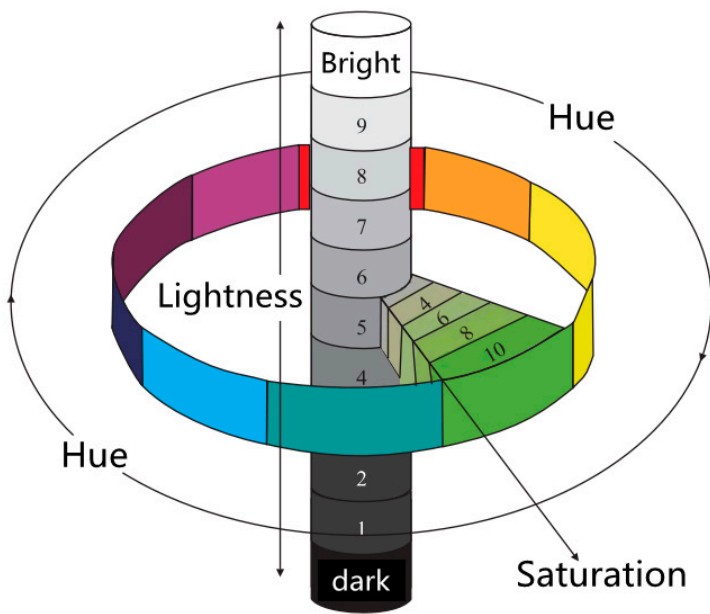

**Figure 3.** Munsell color system.

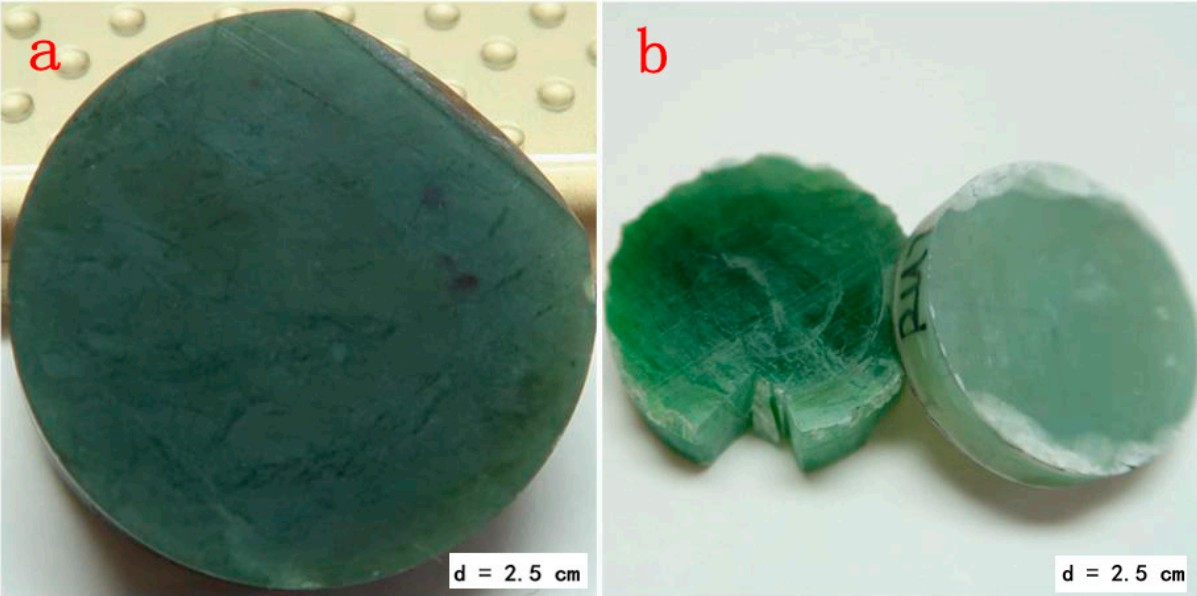

**Figure 4.** Color comparison of green nephrites from Manas (**a**) and East Sayan (**b**).

Many Manas green nephrite samples show an uneven, sugary, light yellow–dark brown color, and have a glassy–greasy luster. They are translucent to almost opaque, and some samples have a fibrous structure. The nephrite is coarse-grained and slightly lineated, with a silky luster observed under reflected light. Some samples show a crumple structure with brown filamentous fillings. Occasionally, white oolitic globules were found to be densely packed, with some white sinuous vein intrusion. Black and green spots can also be seen, and the green ones occur as small clusters with brighter colors than the base. Yellowish–green, highly transparent veins are also visible (Figure 5).

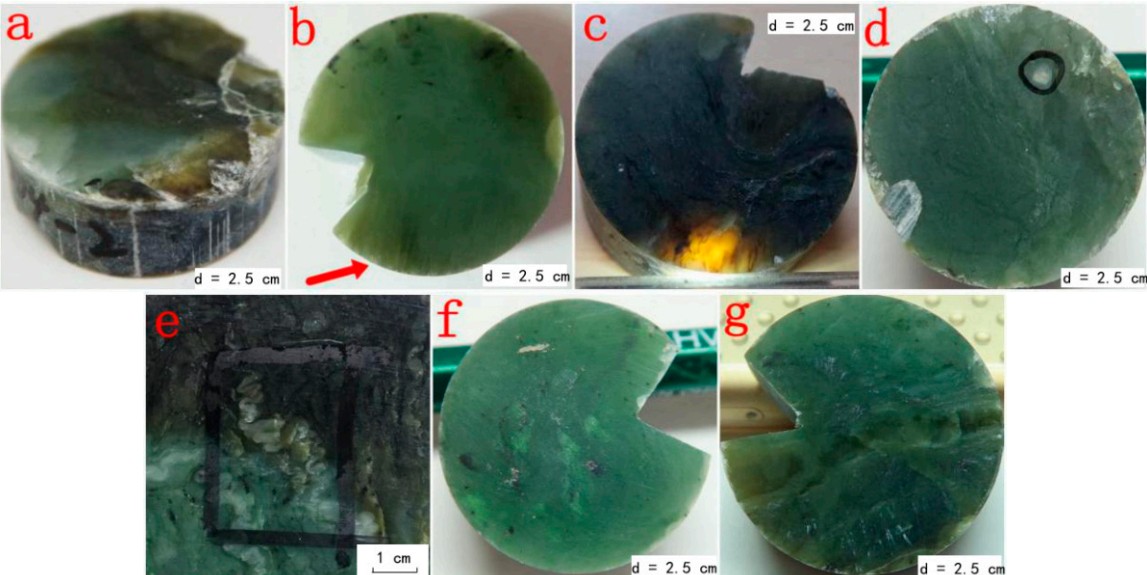

**Figure 5.** Appearance features of the Manas green nephrite samples (**a**). Sugary (**b**). Fibrous (**c**). Crumple and brown filling (**d**). White oolitic globules (**e**). White zigzag veins (**f**). Black and green spots (**g**). Yellowish-green vein.

The East Sayan green nephrite samples have a waxy–glassy luster, notably the emerald-green sample from No. 7 mine. The duck-egg blue sample has a homogenous texture and shows no discernible structure. The samples are translucent to almost opaque (No. 7 mine), with those from Arahushun being more transparent with an ice-like substrate. Black spots are common; those from No. 10 mine samples are more evenly distributed and medium-sized, whilst those from No. 37 mine samples are larger and sparsely distributed. Some black spots on the surface show a metallic luster under reflected light, and those from the Arahushun mine pebble sample are small and dense (Figure 6).

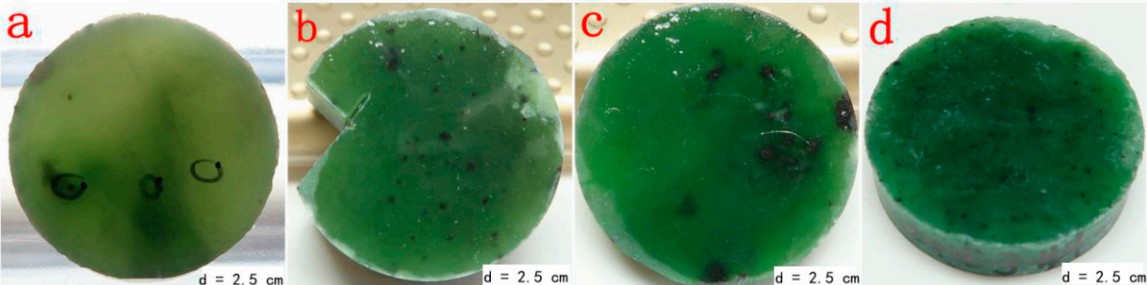

**Figure 6.** Appearance features of the East Sayan green nephrite samples (**a**). Ice-like substrate in Arahushun mine; (**b**). Black spots in No. 10 mine sample; (**c**). Black spots in No. 37 mine sample are large and sparsely-distributed; (**d**). Black spots in Arahushun mine pebble sample are small and dense.

### 3.2. EPMA Results

### 3.2.1. Major Minerals

The EPMA-obtained major element oxide contents of amphiboles are shown in Tables 1 and 2. Their $SiO_2$ content is close to the theoretical values of the general formula $Ca_2(Mg^{2+}, Fe^{2+})_5[Si_4O_{11}]_2(OH)_2$ of tremolite-actinolite: $SiO_2$ (59.17 wt.%), CaO (13.80 wt.%), and belong to the tremolite-actinolite series.

**Table 1.** EPMA major element oxide contents (wt.%) and crystal chemical formula of amphiboles (tremolite) in the Manas green nephrite.

| Sample No. | QKT-02-1-1 | QKT-02-1-2 | QKT-02-1-3 | QKT-02-1-4 | QKT-02-1-5 | QKT-02-1-6 | QKT-02-1-7 | QKT-02-1-8 | QKT-07-1-2 | QKT-07-1-4 | QKT-07-1-6 | QKT-10-1-1 | QKT-10-1-2 | QKT-10-1-3 | QKT-10-1-4 | QKT-10-1-5 | QKT-10-3-1 | QKT-10-3-2 | QKT-10-3-3 | QKT-10-3-4 | QKT-10-3-8 |
|---|---|---|---|---|---|---|---|---|---|---|---|---|---|---|---|---|---|---|---|---|---|
| $Na_2O$ | 0.33 | 0.1 | 0.25 | 0.15 | 0.22 | 0.22 | 0.21 | 0.23 | 0.22 | 0.25 | 0.11 | 0.27 | 0.28 | 0.15 | 0.21 | 0.14 | 0.15 | 0.37 | 0.21 | 0.19 | 0.18 |
| MgO | 21.76 | 19.79 | 21.61 | 21.54 | 21.34 | 21.66 | 21.93 | 21.99 | 20.98 | 21.56 | 21.1 | 22.41 | 21.68 | 21.73 | 21.58 | 21.72 | 21.92 | 22.07 | 21.73 | 21.86 | 21.64 |
| $Al_2O_3$ | 0.18 | 0 | 0.19 | 0.18 | 0.02 | 0.14 | 0.19 | 0.38 | 0.63 | 0.19 | 0.02 | 0.03 | 0.09 | 0.1 | 0 | 0.09 | 0.1 | 0.19 | 0.1 | 0 | 0 |
| $SiO2$ | 57.93 | 57.8 | 58.07 | 58.06 | 58.03 | 58.29 | 58.3 | 57.79 | 57.17 | 57.91 | 57.53 | 57.58 | 58.03 | 58.47 | 58.13 | 58.21 | 58.13 | 57.91 | 57.84 | 57.62 | 57.97 |
| $K_2O$ | 0.06 | 0 | 0.07 | 0.01 | 0 | 0 | 0 | 0.02 | 0.02 | 0 | 0 | 0 | 0.03 | 0.06 | 0.02 | 0 | 0 | 0.03 | 0 | 0.01 | 0 |
| CaO | 12.77 | 12.91 | 12.85 | 13.12 | 13.02 | 13.42 | 13.06 | 12.98 | 13.29 | 13.22 | 13.3 | 12.7 | 12.43 | 12.92 | 13.34 | 13.18 | 13.15 | 12.83 | 13.22 | 13.25 | 12.91 |
| $TiO2$ | 0.16 | 0 | 0.05 | 0 | 0 | 0 | 0 | 0 | 0 | 0.09 | 0 | 0.08 | 0 | 0 | 0 | 0 | 0 | 0 | 0 | 0.07 | 0.04 |
| $Cr_2O_3$ | 0.1 | 0.08 | 0.03 | 0 | 0.06 | 0.01 | 0.04 | 0.14 | 0.24 | 0.01 | 0.08 | 0.07 | 0.06 | 0 | 0.02 | 0.06 | 0.12 | 0.1 | 0.13 | 0 | 0.12 |
| MnO | 0.22 | 0.09 | 0.24 | 0.19 | 0.04 | 0.12 | 0.18 | 0.3 | 0.23 | 0.2 | 0.17 | 0.12 | 0.17 | 0.17 | 0.12 | 0.07 | 0.09 | 0.26 | 0.01 | 0.11 | 0.17 |
| TFeO | 3.49 | 6.27 | 3.53 | 3.64 | 4.12 | 3.39 | 3.57 | 3.37 | 4.06 | 3.73 | 4 | 3.7 | 4.2 | 3.86 | 3.93 | 4.03 | 3.15 | 3.35 | 3.99 | 3.54 | 4.23 |
| NiO | 0.35 | 0.17 | 0.32 | 0.47 | 0.38 | 0.21 | 0.38 | 0.13 | 0.09 | 0.18 | 0.14 | 0.19 | 0.08 | 0.06 | 0.24 | 0.2 | 0.16 | 0.1 | 0.22 | 0.23 | 0.22 |
| ZnO | 0.09 | 0 | 0.12 | 0 | 0.23 | 0.08 | 0 | 0.37 | 0 | 0.3 | 0 | 0.06 | 0.18 | 0 | 0.07 | 0 | 0 | 0.32 | 0.07 | 0.18 | 0 |
| Total | 97.44 | 97.21 | 97.31 | 97.36 | 97.45 | 97.55 | 97.86 | 97.71 | 96.94 | 97.65 | 96.44 | 97.21 | 97.24 | 97.51 | 97.65 | 97.7 | 96.97 | 97.55 | 97.52 | 97.06 | 97.49 |
| Crystal Chemical Formula | | | | | | | | | | | | | | | | | | | | | |
| Si | 8 | 8 | 8 | 8 | 8 | 8 | 8 | 8 | 8 | 8 | 8 | 7.97 | 8 | 8 | 8 | 8 | 8 | 8 | 8 | 8 | 8 |
| Al | 0 | 0 | 0 | 0 | 0 | 0 | 0 | 0 | 0 | 0 | 0 | 0.01 | 0 | 0 | 0 | 0 | 0 | 0 | 0 | 0 | 0 |
| Ti | 0 | 0 | 0 | 0 | 0 | 0 | 0 | 0 | 0 | 0 | 0 | 0.01 | 0 | 0 | 0 | 0 | 0 | 0 | 0 | 0 | 0 |
| $Fe^{3+}$ | 0 | 0 | 0 | 0 | 0 | 0 | 0 | 0 | 0 | 0 | 0 | 0.02 | 0 | 0 | 0 | 0 | 0 | 0 | 0 | 0 | 0 |
| $\sum T$ | 8 | 8 | 8 | 8 | 8 | 8 | 8 | 8 | 8 | 8 | 8 | 8 | 8 | 8 | 8 | 8 | 8 | 8 | 8 | 8 | 8 |
| Al | 0.03 | 0 | 0.03 | 0.03 | 0 | 0.02 | 0.03 | 0.06 | 0.1 | 0.03 | 0 | 0 | 0.02 | 0.02 | 0 | 0.02 | 0.02 | 0.03 | 0.02 | 0 | 0 |
| Ti | 0.02 | 0 | 0.01 | 0 | 0 | 0 | 0 | 0 | 0 | 0.01 | 0 | 0 | 0 | 0 | 0 | 0 | 0 | 0 | 0 | 0.01 | 0 |
| $Cr^{3+}$ | 0.01 | 0.01 | 0 | 0 | 0.01 | 0 | 0 | 0.02 | 0.03 | 0 | 0.01 | 0.01 | 0.07 | 0 | 0 | 0.01 | 0.01 | 0.01 | 0.01 | 0 | 0.01 |
| $Fe^{3+}$ | 0.12 | 0.48 | 0.26 | 0.26 | 0.34 | 0.25 | 0.17 | 0 | 0 | 0.1 | 0.25 | 0.07 | 0.27 | 0.33 | 0.21 | 0.18 | 0.25 | 0.05 | 0.05 | 0.08 | 0.16 |
| Mg | 4.48 | 4.08 | 4.44 | 4.43 | 4.39 | 4.43 | 4.49 | 4.54 | 4.38 | 4.44 | 4.37 | 4.62 | 4.46 | 4.43 | 4.43 | 4.45 | 4.5 | 4.55 | 4.48 | 4.53 | 4.45 |
| Fe | 0.29 | 0.25 | 0.15 | 0.16 | 0.14 | 0.14 | 0.25 | 0.39 | 0.48 | 0.33 | 0.21 | 0.3 | 0.22 | 0.12 | 0.24 | 0.29 | 0.11 | 0.34 | 0.41 | 0.33 | 0.33 |
| Mn | 0.026 | 0.011 | 0.028 | 0.022 | 0.005 | 0.014 | 0.021 | 0 | 0.02 | 0.023 | 0.02 | 0 | 0.02 | 0.02 | 0.014 | 0.008 | 0.01 | 0.026 | 0.001 | 0.013 | 0.02 |
| $\sum C$ | 4.965 | 4.828 | 4.912 | 4.895 | 4.875 | 4.858 | 4.952 | 5 | 5 | 4.936 | 4.871 | 5 | 4.981 | 4.91 | 4.896 | 4.942 | 4.899 | 5 | 4.974 | 4.956 | 4.977 |
| Mg | 0 | 0 | 0 | 0 | 0 | 0 | 0 | 0 | 0 | 0 | 0 | 0 | 0 | 0 | 0 | 0 | 0 | 0 | 0 | 0 | 0 |
| Fe | 0 | 0 | 0 | 0 | 0 | 0 | 0 | 0 | 0 | 0 | 0 | 0.04 | 0 | 0 | 0 | 0 | 0 | 0 | 0 | 0 | 0 |
| Mn | 0 | 0 | 0 | 0 | 0 | 0 | 0 | 0.035 | 0 | 0 | 0 | 0.014 | 0 | 0 | 0 | 0 | 0 | 0.004 | 0 | 0 | 0 |
| Ca | 1.89 | 1.92 | 1.9 | 1.94 | 1.92 | 1.97 | 1.92 | 1.92 | 1.99 | 1.96 | 1.98 | 1.88 | 1.84 | 1.89 | 1.97 | 1.94 | 1.94 | 1.9 | 1.96 | 1.97 | 1.91 |
| Na | 0.09 | 0.03 | 0.07 | 0.04 | 0.06 | 0.03 | 0.06 | 0.04 | 0 | 0.04 | 0.02 | 0.06 | 0.08 | 0.04 | 0.03 | 0.04 | 0.04 | 0.1 | 0.04 | 0.03 | 0.05 |
| $\sum B$ | 1.98 | 1.94 | 1.96 | 1.98 | 1.98 | 2 | 1.98 | 2 | 2 | 2 | 2 | 2 | 1.91 | 1.93 | 2 | 1.98 | 1.98 | 2 | 2 | 2 | 1.96 |
| Ca | 0 | 0 | 0 | 0 | 0 | 0 | 0 | 0 | 0 | 0 | 0 | 0 | 0 | 0 | 0 | 0 | 0 | 0 | 0 | 0 | 0 |
| Na | 0 | 0 | 0 | 0 | 0 | 0.03 | 0 | 0.02 | 0.06 | 0.02 | 0.01 | 0.01 | 0 | 0 | 0.02 | 0 | 0 | 0 | 0.02 | 0.02 | 0 |
| K | 0.01 | 0 | 0.01 | 0 | 0 | 0 | 0 | 0 | 0 | 0 | 0 | 0 | 0.01 | 0.01 | 0 | 0 | 0 | 0.01 | 0 | 0 | 0 |
| $\sum A$ | 0.01 | 0 | 0.01 | 0 | 0 | 0.03 | 0 | 0.03 | 0.06 | 0.02 | 0.01 | 0.01 | 0.01 | 0.01 | 0.03 | 0 | 0 | 0.01 | 0.02 | 0.02 | 0 |
| $Mg/(Mg+Fe^{2+})$ | 0.94 | 0.94 | 0.97 | 0.97 | 0.97 | 0.97 | 0.95 | 0.92 | 0.9 | 0.93 | 0.95 | 0.94 | 0.95 | 0.97 | 0.95 | 0.94 | 0.98 | 0.93 | 0.92 | 0.93 | 0.93 |

Note: $TFeO = FeO + Fe_2O_3$.

**Table 2.** EPMA data of major element oxides (wt.%) and crystal chemical formula of amphiboles (tremolite and actinolite) in the East Sayan green nephrite samples.

| Sample No. | RU-Z-1-1 | RU-Z-1-3 | RU7-01-1-1 | RU7-01-1-2 | RU7-02-1-1 | RU7-02-1-2 | RU7-03-1-1 | RU7-03-1-2 | RU7-03-1-4 | RU37-02-2-1 | RU37-02-2-2 | RU37-02-2-3 | RU37-02-2-4 | RU37-02-2-5 |
|---|---|---|---|---|---|---|---|---|---|---|---|---|---|---|
| $Na_2O$ | 0.32 | 0.24 | 0.17 | 0.26 | 0.23 | 0.24 | 0.26 | 0.34 | 0.29 | 0.25 | 0.34 | 0.25 | 0.29 | 0.39 |
| MgO | 21.6 | 21.53 | 21.76 | 21.93 | 21.14 | 21.79 | 21.95 | 21.42 | 21.9 | 21.31 | 20.43 | 20.89 | 19.43 | 19.91 |
| $Al_2O_3$ | 0.23 | 0.15 | 0.2 | 0.22 | 0.16 | 0.26 | 0.15 | 0.26 | 0.17 | 0.23 | 0.39 | 0.26 | 0.21 | 0.2 |
| $SiO_2$ | 57.24 | 58.02 | 58.54 | 58.68 | 57.69 | 57.89 | 58.55 | 57.74 | 58.27 | 57.45 | 56.98 | 57.31 | 56.65 | 57.08 |
| $K_2O$ | 0.05 | 0.03 | 0 | 0 | 0 | 0.05 | 0.07 | 0.06 | 0 | 0.03 | 0 | 0.04 | 0 | 0 |
| CaO | 13.33 | 13.44 | 12.74 | 12.69 | 12.81 | 12.38 | 12.61 | 12.79 | 12.4 | 13.13 | 13.36 | 13.42 | 13.12 | 13.44 |
| $TiO_2$ | 0.1 | 0.01 | 0 | 0.01 | 0 | 0 | 0.1 | 0.02 | 0.03 | 0.12 | 0.03 | 0.01 | 0.06 | 0.1 |
| $Cr_2O_3$ | 0.45 | 0.25 | 0.03 | 0.18 | 0.04 | 0 | 0.12 | 0.29 | 0.16 | 0.4 | 0.22 | 0.21 | 0.21 | 0.15 |
| MnO | 0.2 | 0.23 | 0.23 | 0.01 | 0.2 | 0.14 | 0.08 | 0.08 | 0.2 | 0.15 | 0.13 | 0.18 | 0.15 | 0.16 |
| TFeO | 2.76 | 2.81 | 3.63 | 3.64 | 4.07 | 4.3 | 3.6 | 3.83 | 3.4 | 3.48 | 5.14 | 4 | 6.91 | 6.33 |
| NiO | 0.3 | 0.33 | 0.36 | 0 | 0.05 | 0 | 0 | 0.23 | 0.28 | 0.34 | 0.34 | 0.15 | 0.29 | 0.25 |
| ZnO | 0.11 | 0.15 | 0 | 0.05 | 0.04 | 0.33 | 0.27 | 0.35 | 0.11 | 0.28 | 0.23 | 0.07 | 0.1 | 0.12 |
| Total | 96.69 | 97.17 | 97.66 | 97.67 | 96.42 | 97.37 | 98 | 97.46 | 97.27 | 97.17 | 97.4 | 96.93 | 97.38 | 97.87 |
| Crystal Chemical Formula | | | | | | | | | | | | | | |
| Si | 8 | 8 | 8 | 8 | 8 | 8 | 8 | 8 | 8 | 8 | 8 | 8 | 8 | 8 |
| Al | 0 | 0 | 0 | 0 | 0 | 0 | 0 | 0 | 0 | 0 | 0 | 0 | 0 | 0 |
| Ti | 0 | 0 | 0 | 0 | 0 | 0 | 0 | 0 | 0 | 0 | 0 | 0 | 0 | 0 |
| $Fe^{3+}$ | 0 | 0 | 0 | 0 | 0 | 0 | 0 | 0 | 0 | 0 | 0 | 0 | 0 | 0 |
| $\sum T$ | 8 | 8 | 8 | 8 | 8 | 8 | 8 | 8 | 8 | 8 | 8 | 8 | 8 | 8 |
| Al | 0.04 | 0.02 | 0.03 | 0.04 | 0.03 | 0.04 | 0.02 | 0.04 | 0.03 | 0.04 | 0.07 | 0.04 | 0.04 | 0.03 |
| Ti | 0.01 | 0 | 0 | 0 | 0 | 0 | 0.01 | 0 | 0 | 0.01 | 0 | 0 | 0.01 | 0.01 |
| $Cr^{3+}$ | 0.05 | 0.03 | 0 | 0.02 | 0 | 0 | 0.01 | 0.03 | 0.02 | 0.04 | 0.02 | 0.02 | 0.02 | 0.02 |
| $Mn^{3+}$ | 0 | 0 | 0 | 0 | 0 | 0 | 0 | 0 | 0 | 0 | 0 | 0 | 0 | 0 |
| $Fe^{3+}$ | 0 | 0.25 | 0.37 | 0.31 | 0.31 | 0.13 | 0.29 | 0.12 | 0.34 | 0.02 | 0 | 0.04 | 0 | 0 |
| Mg | 4.5 | 4.43 | 4.43 | 4.46 | 4.37 | 4.49 | 4.47 | 4.42 | 4.48 | 4.42 | 4.28 | 4.35 | 4.09 | 4.16 |
| Fe | 0.32 | 0.08 | 0.05 | 0.11 | 0.16 | 0.34 | 0.12 | 0.33 | 0.06 | 0.39 | 0.6 | 0.43 | 0.82 | 0.74 |
| Mn | 0.02 | 0.03 | 0.03 | 0 | 0.02 | 0 | 0.01 | 0.01 | 0.02 | 0.02 | 0.02 | 0.02 | 0.02 | 0.02 |
| Li | 0 | 0 | 0 | 0 | 0 | 0 | 0 | 0 | 0 | 0 | 0 | 0 | 0 | 0 |
| $\sum C$ | 4.95 | 4.83 | 4.91 | 4.93 | 4.9 | 5 | 4.94 | 4.95 | 4.94 | 4.94 | 4.99 | 4.9 | 4.99 | 4.98 |
| Fe | 0 | 0 | 0 | 0 | 0 | 0.03 | 0 | 0 | 0 | 0 | 0 | 0 | 0 | 0 |
| Mn | 0 | 0 | 0 | 0 | 0 | 0.016 | 0 | 0 | 0 | 0 | 0 | 0 | 0 | 0 |
| Li | 0 | 0 | 0 | 0 | 0 | 0 | 0 | 0 | 0 | 0 | 0 | 0 | 0 | 0 |
| Ca | 2 | 1.99 | 1.87 | 1.85 | 1.9 | 1.83 | 1.85 | 1.9 | 1.82 | 1.96 | 2 | 2 | 1.99 | 2 |
| Na | 0 | 0 | 0.05 | 0.07 | 0.06 | 0.06 | 0.07 | 0.09 | 0.08 | 0.04 | 0.04 | 0.04 | 0.02 | 0.02 |
| $\sum B$ | 2 | 2 | 1.91 | 1.92 | 1.97 | 1.94 | 1.92 | 1.99 | 1.9 | 2 | 2.04 | 2.04 | 2 | 2.02 |
| Ca | 0 | 0 | 0 | 0 | 0 | 0 | 0 | 0 | 0 | 0 | 0 | 0.01 | 0.01 | 0.02 |
| Na | 0.08 | 0.05 | 0 | 0 | 0 | 0 | 0 | 0 | 0 | 0 | 0.03 | 0.09 | 0.07 | 0.11 |
| K | 0.01 | 0.01 | 0 | 0 | 0 | 0.01 | 0.01 | 0.01 | 0 | 0 | 0.01 | 0 | 0.01 | 0 |
| $\sum A$ | 0.09 | 0.06 | 0 | 0 | 0 | 0.01 | 0.01 | 0.01 | 0 | 0.03 | 0.1 | 0.08 | 0.07 | 0.12 |
| $Mg/(Mg + Fe^{2+})$ | 0.93 | 0.98 | 0.99 | 0.98 | 0.96 | 0.92 | 0.97 | 0.93 | 0.99 | 0.92 | 0.88 | 0.91 | 0.83 | 0.85 |
| Mineral | Tre. | Tre. | Tre. | Tre. | Tre. | Tre. | Tre. | Tre. | Tre. | Tre. | Act. | Tre. | Act. | Act. |

Note: 1. TfeO = FeO + $Fe_2O_3$; 2. Tre. Is short for tremolite. Act. Is short for actinolite.

Amphiboles in Manas samples have a lower $Cr_2O_3$ content (0–0.24%, avg. 0.08%) than that of the East Sayan samples (0–0.45%, avg. 0.21%). Moreover, they have a low Ti content, mostly below the detection limit. To explore the relationship among diffident chemical components of green nephrites from different origins, binary plots of $SiO_2$-($Na_2O$ + $K_2O$), TfeO-($Na_2O$ + $K_2O$) and MgO-TfeO were constructed (Figure 7). Amphiboles in Manas and East Sayan samples fall into a similar area in the $SiO_2$-($Na_2O$ + $K_2O$) and TfeO-($Na_2O$ + $K_2O$) plots. In the MgO-TfeO plot, they show negative MgO vs. FeO correlations, suggesting an isomorphic substitution between Mg and Fe in tremolite.

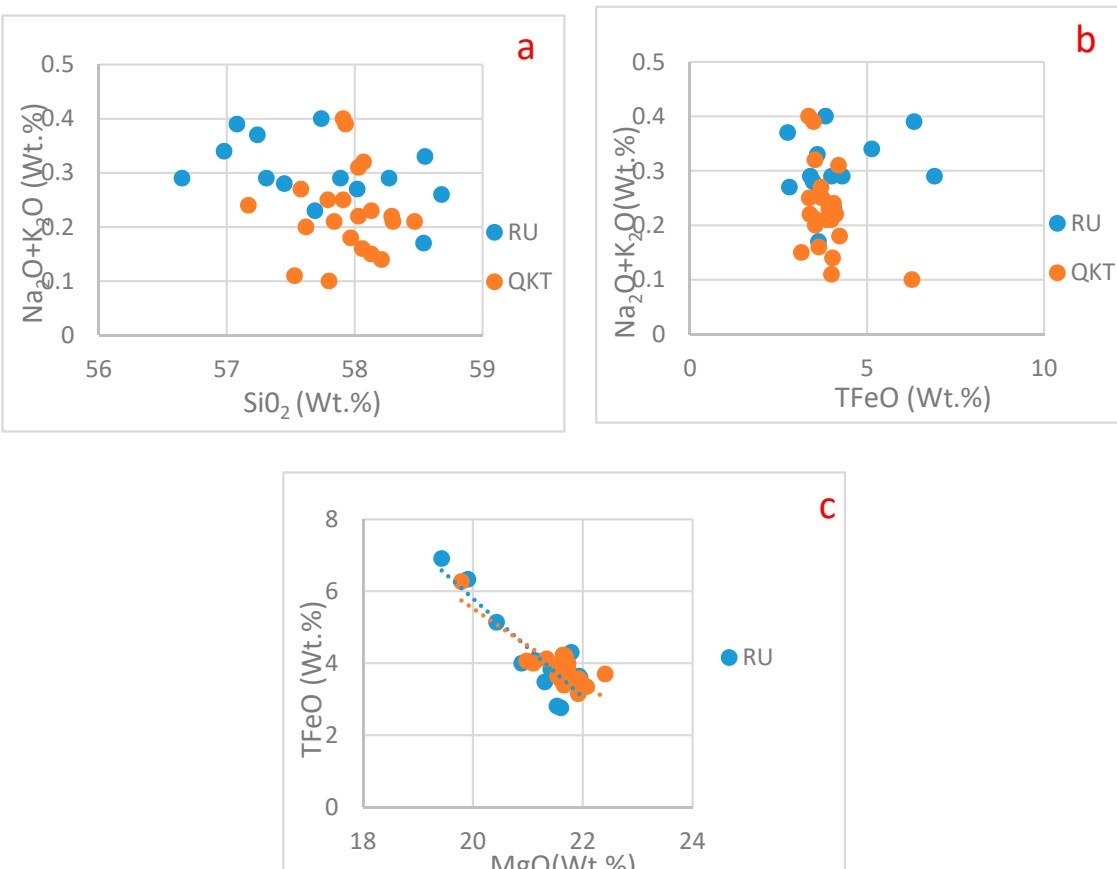

**Figure 7.** Binary plots of (**a**) $SiO_2$-($Na_2O$ + $K_2O$), (**b**) TfeO-($Na_2O$ + $K_2O$) and (**c**) MgO-TfeO for the Manas and East Sayan green nephrite samples (TFeO = FeO + $Fe_2O_3$).

According to the International Mineralogical Association (IMA) naming scheme for amphiboles [40], the cationic coefficients (including $Fe^{2+}$ and $Fe^{3+}$) in the oxide were calculated based on 23 oxygen ions [41], and their crystal chemical formula was determined (Tables 1 and 2).

According to the amphibole Nomenclature Scheme of the IMA Commission on New Minerals, Nomenclature and Classification (CNMNC) [42], all the analyzed samples have $(Ca + Na)_B \geq 1.34$, $Na_B < 0.67$, $Ca_B > 1.34$, and $Si \geq 7.95$, belonging to calcareous amphibolite. Tremolite and actinolite are classified according to $Mg/(Mg + Fe^{2+})$ (tremolite: 0.90–1.00; actinolite = 0.50–0.90; ferroactinolite = 0.00–0.50), and both the Manas and East Sayan samples belong to tremolite (with minor actinolite). Among these samples, those from Arahushun mine pebbles and No. 7 mine have a relatively high $Mg/(Mg + Fe^{2+})$ (>0.95), whereas those from No. 37 mine have a relatively low $Mg/(Mg + Fe^{2+})$ (<0.9), indicating that the latter samples are actinolite (Tables 1 and 2).

### 3.2.2. Impurities (Minor Minerals)

Minor component minerals (impurities) are common in green nephrites. Chlorite-group minerals, uvarovite and chromite were found as impurity minerals in the Manas samples, whilst chlorite-group minerals, chromite and bornite were found in the East Sayan samples. The BSE images of impurities are shown in Figure 8. The EPMA data of these mineral inclusions are shown in Table 3.

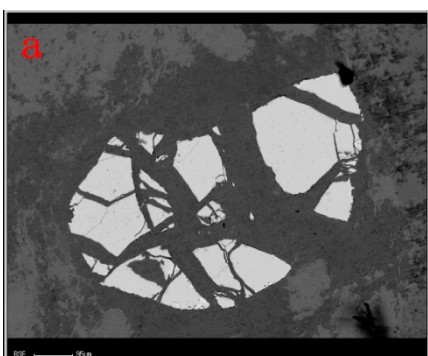 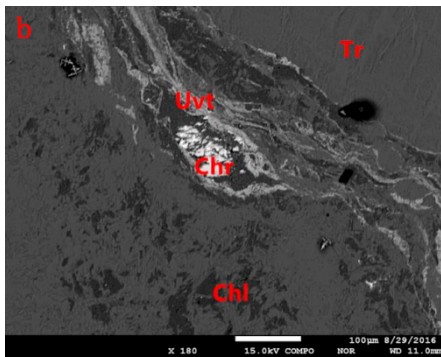 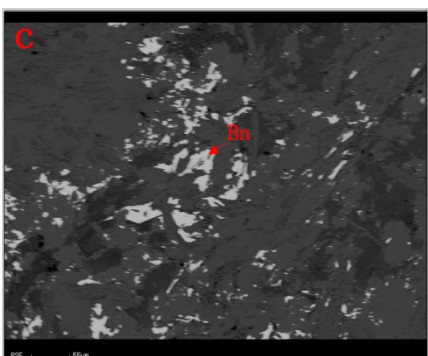

**Figure 8.** BSE images of (**a**) chromite in Manas green nephrite, (**b**) chlorite-group minerals (Chl), uvarovite (Uvt) and chromite (Chr) in Manas green nephrite, and (**c**) bornite (Bn) in East Sayan green nephrite.

### 3.3. LA-ICP-MS

The trace element contents of twenty-three Manas and 34 East Sayan samples were measured (on the amphibole crystals), and the results are listed in Table 4. Comparing their contents from Manas and East Sayan, the former clearly has a higher Sr (Manas: 13.74–35.56 (avg. 22.75) ppm; East Sayan: 2.87–9.68 (avg. 5.72) ppm) but lower Cr (Manas: 8.93–321.61 (avg. 81.91) ppm; East Sayan: 11.94–1797.49 (avg. 616.26) ppm), Zn (Manas: 23.43–52.73 (avg. 31.13) ppm; East Sayan: 27.53–149.42 (avg. 104.34) ppm), Y (Manas: 0.01–0.09 (avg. 0.030) ppm; East Sayan: 0.16–0.63 (avg. 0.33) ppm) and Ba (Manas: 0.31–1.35 (avg. 0.61) ppm; East Sayan: 0.57–16.31 (avg. 4.30) ppm) contents.

In the binary Cr-V, Sr-Y and Sr-Ba plots (Figure 9), the Manas and East Sayan data points fall in distinct fields. In the Cr-V plot, the Manas samples have a narrower range and lower Cr and V contents than the East Sayan samples. In the Sr-Y plot, the Manas samples contain a higher Sr but lower Y than the East Sayan samples. In the Sr-Ba plot, the data point distributions of the Manas and East Sayan samples are similar to those in the Sr-Y plot, and certain positive Sr vs. Ba correlations can be found (with different correlation coefficients). The slope of the Manas samples is gentler than that of the East Sayan samples.

In the chondrite-normalized rare earth element (REE) plots (Figure 10), amphiboles in the Manas samples have mostly positive anomalies for europium ($\delta$Eu) and cerium ($\delta$Ce, QKT-02-1-7 sample), whereas those in the East Sayan samples have mostly negative $\delta$Eu (except for the pebble nephrite samples) and positive $\delta$Ce (except for samples RU7#02-1-2 and RU7#03-1-4) values.

Amphiboles in the Manas samples have a lower total ($\sum$)REE, light REE (LREE) and heavy REE (HREE) contents, but a higher $\sum$LREE/$\sum$HREE ratio than the East Sayan samples. The REE distribution patterns of the Manas are right-inclining (LREE/HREE-enriched). Most of the East Sayan samples have right-inclining (though gentler) REE patterns, and a few exhibit a U-shape (relative middle REE (MREE) depletion). In addition, the $(La/Yb)_N$, $(La/Sm)_N$, $(Gd/Yb)_N$ ratios of the Manas samples are generally higher than those of the East Sayan samples, indicating that the former are more fractionated. The Eu/Sm values of the Manas samples are generally higher than those of the East Sayan samples (Table 5).

**Table 3.** EPMA data of mineral inclusions in the Manas and East Sayan green nephrites (wt.%).

| Sample No. | QKT-07-1-1 | QKT-07-1-3 | QKT-07-1-5 | QKT-02-1-8 | QKT-10-3-1 | RU-Z-1-2 | RU-Z-1-3 | RU-Z-1-4 | QKT-02-1-8 | QKT-10-3-1 | RU-Z-1-1 | RU37-02-2-2 | QKT-01-1-3 | QKT-02-2-1-5 | Sample No. | RU-Z-1-1 |
|---|---|---|---|---|---|---|---|---|---|---|---|---|---|---|---|---|
| $Na_2O$ | 0.22 | 0.25 | 0.29 | 0.42 | 0.38 | 0.48 | 0.34 | 0.38 | 0.39 | 0.33 | 0.30 | 0.35 | 0.01 | 0 | S | 25.73 |
| MgO | 11.52 | 11.20 | 11.36 | 11.18 | 10.45 | 1.61 | 3.86 | 2.65 | 29.37 | 28.73 | 25.93 | 26.40 | 4.17 | 0.92 | Fe | 8.81 |
| $Al_2O_3$ | 15.36 | 15.53 | 16.75 | 15.69 | 13.97 | 15.86 | 15.81 | 15.83 | 15.14 | 15.06 | 16.56 | 15.91 | 1.49 | 0.71 | Co | 0.09 |
| $SiO_2$ | 0.25 | 0.17 | 0.20 | 0.18 | 0.21 | 0.15 | 0.18 | 0.12 | 34.02 | 32.86 | 31.70 | 32.59 | 38.31 | 34.87 | Cu | 63.57 |
| $K_2O$ | 0 | 0.03 | 0 | 0 | 0.05 | 0 | 0.08 | 0 | 0 | 0 | 0 | 0 | 0.02 | 0 | As | 1.03 |
| CaO | 0.09 | 0.06 | 0.08 | 0.07 | 0.08 | 0.08 | 0.03 | 0.03 | 0.13 | 0.09 | 0.97 | 0.50 | 28.53 | 30.73 | Total | 99.23 |
| $TiO_2$ | 0.15 | 0.05 | 0.09 | 0.17 | 0.13 | 0.15 | 0.05 | 0.09 | 0.02 | 0.01 | 0 | 0 | 0.22 | 0.08 | Mineral | Bor. |
| $Cr_2O_3$ | 52.21 | 51.99 | 51.00 | 51.94 | 53.26 | 46.79 | 47.29 | 47.38 | 1.05 | 1.76 | 3.30 | 1.70 | 13.92 | 20.74 | | |
| MnO | 0.82 | 0.43 | 0.48 | 0.34 | 0.39 | 6.52 | 5.13 | 5.77 | 0.24 | 0.16 | 0.38 | 0.27 | 0.08 | 0.13 | | |
| FeO | 18.19 | 19.45 | 18.75 | 19.80 | 19.51 | 26.92 | 25.88 | 26.89 | 6.22 | 6.39 | 7.05 | 8.30 | 6.56 | 3.72 | | |
| NiO | 0.15 | 0 | 0.23 | 0.04 | 0.12 | 0.12 | 0.13 | 0.17 | 0 | 0 | 0 | 0 | 0 | 0 | | |
| ZnO | 0.02 | 0.35 | 0.62 | 0.01 | 0.13 | 1.53 | 0.62 | 0.80 | 0.20 | 0.23 | 0 | 0 | 0 | 0 | | |
| Total | 98.99 | 99.50 | 99.85 | 99.82 | 98.67 | 100.16 | 99.39 | 100.11 | 86.78 | 85.61 | 86.18 | 86.03 | 94.23 | 94.46 | | |
| Mineral | Chr. | Chr. | Chr. | Chr. | Chr. | Chr. | Chr. | Chr. | Chl. | Chl. | Chl. | Chl. | Uva. | Uva. | | |

Note: Chr = chromite; Chl = chlorite-group minerals; Uva = uvarovite; Bor = bornite.

**Table 4.** Representative LA-ICP-MS data of amphiboles in the Manas and East Sayan green nephrite (ppm).

| Sample No. | RU7#-02-1-1 | RU7#01-1-1 | RU7#03-1-1 | RU-ZL-1-5 | RU-ZL-1-6 | RU37#-02-2-1 | RU37#-02-2-2 | QKT-10-3-1 | QKT-10-3-2 | QKT-02-1-1 | QKT-02-1-2 | QKT-10-1-1 | QKT-10-1-2 |
|---|---|---|---|---|---|---|---|---|---|---|---|---|---|
| Li | 0.675 | 0.57 | 0.57 | 0.845 | 0.704 | 0.63 | 1.1 | 1.96 | 1.53 | 0.51 | 1.01 | 2.33 | 2.36 |
| Be | 0.441 | 0.58 | 0.52 | 0.685 | 0.526 | 0.68 | 0.67 | 2.4 | 2.75 | 0.91 | 1.04 | 0.56 | 0.71 |
| B | 8.96 | 6.76 | 9.81 | 5.95 | 7.62 | 2.32 | 2.16 | 9.25 | 9.06 | 4.38 | 4.62 | 1.55 | 3.31 |
| P | 8.30 | 8.68 | 7.77 | 8.84 | 7.30 | 5.70 | 6.63 | 4.01 | 3.29 | 9.56 | 9.60 | 13.95 | 16.82 |
| K | 230 | 237 | 383 | 150 | 114 | 201 | 229 | 216 | 177 | 179 | 206 | 224 | 232 |
| Sc | 0.97 | 4.36 | 3.15 | 1.917 | 1.245 | 4.08 | 2.53 | 39.07 | 30.97 | 1.82 | 0.98 | 3.06 | 3.19 |
| Ti | 9.31 | 15.05 | 8.88 | 7.39 | 8.63 | 110.42 | 110.43 | 26.57 | 18.22 | 12.44 | 9.22 | 18.81 | 22.69 |
| V | 7.97 | 11.46 | 9.96 | 7.61 | 8.26 | 61.30 | 63.68 | 4.93 | 3.24 | 5.21 | 3.64 | 11.64 | 16.13 |
| Cr | 13.70 | 538 | 802 | 1081 | 915 | 538 | 422 | 190 | 92.11 | 43.2 | 47.63 | 322 | 134 |
| Mn | 869 | 918 | 845 | 1715 | 1700 | 689 | 703 | 1484 | 1563 | 824 | 870 | 1140 | 1117 |
| Co | 53.53 | 52.70 | 55.45 | 72.03 | 73.06 | 43.50 | 48.87 | 61.27 | 57.54 | 62.43 | 58.77 | 74.65 | 76.45 |
| Ni | 715 | 1072 | 1289 | 1508 | 1762 | 735 | 882 | 1311 | 1310 | 1382 | 1260 | 1147 | 990 |
| Cu | 0.013 | 0.015 | 0.03 | 21.51 | 0.147 | 0.018 | 0.196 | 0.139 | 0.148 | 0.018 | 0.015 | 0.25 | 0.26 |
| Zn | 145 | 137 | 138 | 74.30 | 27.53 | 48.08 | 60.52 | 34.59 | 39.15 | 29.46 | 34.77 | 25.75 | 23.43 |
| Ga | 0.55 | 0.61 | 0.60 | 0.52 | 0.45 | 0.67 | 1.70 | 0.83 | 0.84 | 0.57 | 0.72 | 0.49 | 0.46 |
| Rb | 1.41 | 1.09 | 2.44 | 0.466 | 0.35 | 0.64 | 0.56 | 0.81 | 0.51 | 1.13 | 1.31 | 1.4 | 1.51 |
| Sr | 5.02 | 4.35 | 5.35 | 3.19 | 3.85 | 8.60 | 8.62 | 14.73 | 13.74 | 21.93 | 21.38 | 26.89 | 29.50 |
| Y | 0.185 | 0.305 | 0.181 | 0.552 | 0.576 | 0.50 | 0.51 | 0.009 | 0.011 | 0.015 | 0.015 | 0.069 | 0.026 |

Table 4. *Cont.*

| Sample No. | RU7#-02-1-1 | RU7#01-1-1 | RU7#03-1-1 | RU-ZL-1-5 | RU-ZL-1-6 | RU37#-02-2-1 | RU37#-02-2-2 | QKT-10-3-1 | QKT-10-3-2 | QKT-02-1-1 | QKT-02-1-2 | QKT-10-1-1 | QKT-10-1-2 |
|---|---|---|---|---|---|---|---|---|---|---|---|---|---|
| Zr | 0.19 | 0.74 | 0.269 | 0.031 | 0.006 | 0.80 | 0.46 | 0.006 | 0.003 | 0.008 | 0.007 | 0.160 | 0.170 |
| Nb | 0.046 | 0.067 | 0.069 | 0.067 | 0.063 | 0.158 | 0.19 | 0 | 0.001 | 0.007 | 0.007 | 0.044 | 0.017 |
| Cs | 0.287 | 0.307 | 0.42 | 0.226 | 0.209 | 0.235 | 0.157 | 0.389 | 0.263 | 0.712 | 0.77 | 0.68 | 0.89 |
| Ba | 3.39 | 3.84 | 4.29 | 1.045 | 0.573 | 3.57 | 16.31 | 0.305 | 0.313 | 0.57 | 0.57 | 0.31 | 0.82 |
| La | 0.102 | 0.081 | 0.10 | 0.178 | 0.137 | 0.09 | 0.13 | 0.42 | 0.058 | 0.049 | 0.057 | 0.069 | 0.14 |
| Ce | 0.317 | 0.42 | 0.43 | 0.545 | 0.308 | 0.34 | 0.28 | 18.56 | 0.21 | 0.13 | 0.18 | 0.26 | 0.18 |
| Pr | 0.046 | 0.059 | 0.04 | 0.043 | 0.032 | 0.028 | 0.035 | 0.124 | 0.013 | 0.009 | 0.011 | 0.049 | 0.015 |
| Nd | 0.22 | 0.261 | 0.175 | 0.168 | 0.156 | 0.139 | 0.154 | 0.062 | 0.02 | 0.023 | 0.025 | 0.099 | 0.094 |
| Sm | 0.046 | 0.059 | 0.036 | 0.044 | 0.039 | 0.061 | 0.053 | 0.004 | 0.002 | 0.002 | 0.003 | 0 | 0.008 |
| Eu | 0.003 | 0.002 | 0.004 | 0.022 | 0.026 | 0.002 | 0.004 | 0.012 | 0.007 | 0.004 | 0.006 | 0.040 | 0.018 |
| Gd | 0.028 | 0.056 | 0.021 | 0.037 | 0.051 | 0.041 | 0.038 | 0.36 | 0.027 | 0.01 | 0.006 | 0.037 | 0.007 |
| Tb | 0.004 | 0.006 | 0.004 | 0.005 | 0.006 | 0.004 | 0.01 | 0.001 | 0.001 | 0.001 | 0 | 0.02 | 0.014 |
| Dy | 0.026 | 0.048 | 0.024 | 0.048 | 0.051 | 0.07 | 0.069 | 0.004 | 0.001 | 0.001 | 0.001 | 0.061 | 0.005 |
| Ho | 0.005 | 0.010 | 0.006 | 0.009 | 0.012 | 0.012 | 0.015 | 0.001 | 0.001 | 0.001 | 0.001 | 0.026 | 0.001 |
| Er | 0.02 | 0.027 | 0.017 | 0.031 | 0.032 | 0.048 | 0.047 | 0.002 | 0.002 | 0.001 | 0.001 | 0.046 | 0.016 |
| Tm | 0.003 | 0.005 | 0.003 | 0.002 | 0.003 | 0.004 | 0.009 | 0.001 | 0 | 0.001 | 0 | 0.014 | 0.001 |
| Yb | 0.019 | 0.037 | 0.021 | 0.018 | 0.013 | 0.076 | 0.089 | 0.003 | 0.002 | 0 | 0.002 | 0.066 | 0.038 |
| Lu | 0.004 | 0.007 | 0.003 | 0.002 | 0.001 | 0.011 | 0.018 | 0.001 | 0 | 0.001 | 0.001 | 0.021 | 0.003 |
| Hf | 0.004 | 0.026 | 0.008 | 0.002 | 0 | 0.014 | 0.012 | 0.002 | 0 | 0.003 | 0 | 0.053 | 0 |
| Ta | 0.004 | 0.006 | 0.003 | 0.002 | 0.001 | 0.002 | 0.003 | 0.001 | 0 | 0.001 | 0.001 | 0.031 | 0.02 |
| Th | 0.019 | 0.035 | 0.045 | 0.001 | 0.002 | 0.026 | 0.038 | 0.001 | 0 | 0.001 | 0 | 0.038 | 0 |
| U | 0.011 | 0.026 | 0.056 | 0.021 | 0.019 | 0.064 | 0.061 | 0 | 0 | 0.008 | 0 | 0.022 | 0 |

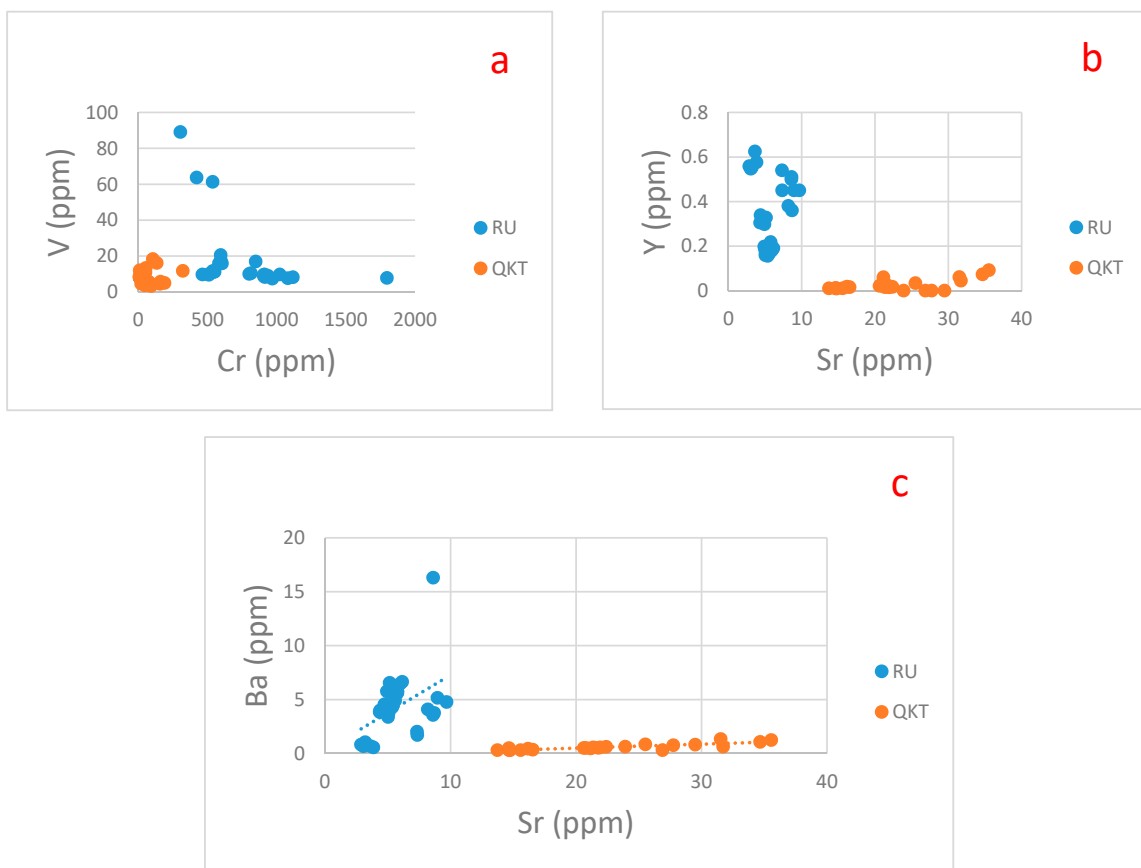

**Figure 9.** Binary (**a**) Cr-V, (**b**) Sr-Y and (**c**) Sr-Ba scatter plots for amphiboles in the Manas and East Sayan green nephrites.

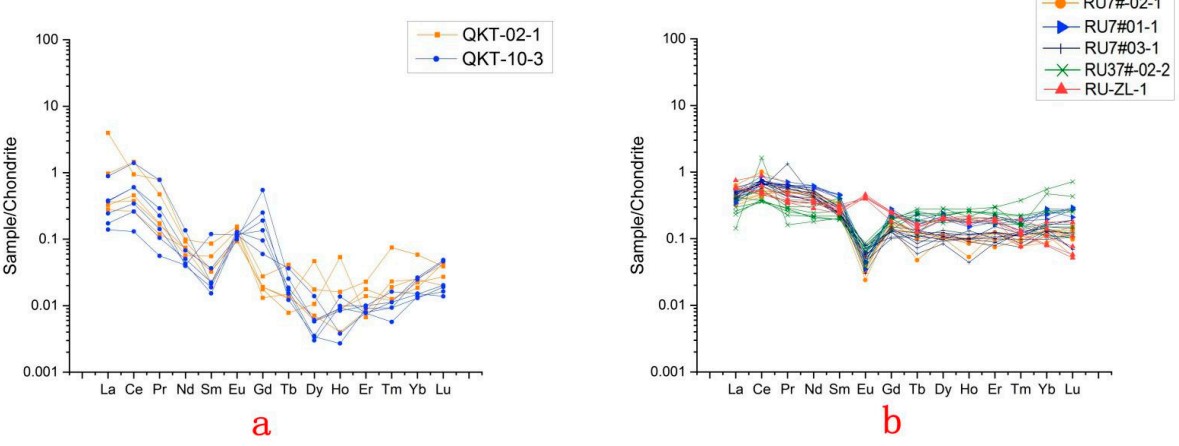

**Figure 10.** Chondrite-normalized REE patterns of amphiboles in the (**a**) Manas and (**b**) East Sayan green nephrite samples.

**Table 5.** REE parameters (average) of the green nephrite samples from Manas and East Sayan.

| Origins | ∑REE (ppm) | ∑LREE (ppm) | ∑HREE (ppm) | Type | ∑LREE/∑HREE | (La/Yb)$_N$ | (La/Sm)$_N$ | (Gd/Yb)$_N$ | δEu | δCe | Eu/Sm | Sm/Nd |
|---------|-----------|-------------|-------------|------|-------------|-------------|-------------|-------------|-----|-----|-------|-------|
| QKT | 0.664 | 0.63 | 0.04 | LREE-rich | 23.77 | 29.17 | 14.744 | 8.463 | Mostly positive | 91% positive | 1.552 | 0.231 |
| RU | 1.194 | 1.04 | 0.16 | LREE-rich (mostly) | 6.46 | 3.24 | 1.617 | 1.249 | Mostly negative (except for pebble nephrite samples) | 97% positive | 0.150 | 0.245 |

Amphiboles in the Manas and East Sayan samples have different REE parameters (Figure 11), and these parameters can be used to distinguish their different origins. By casting our data onto the three-dimensional $\delta Ce$-$\sum REE$-$\sum LREE/\sum HREE$ cartesian coordinate diagram (Figure 12), it can be seen that the data points of the East Sayan samples are relatively clustered, while those from different origins are relatively dispersed, showing a good differentiation. The negative $\delta Eu$ anomaly is usually the result of plagioclase fractionation, and the degree of anomaly increases with magma differentiation. Most of the East Sayan samples have a strongly negative Eu anomaly, indicating that their nephrite-forming fluid may have derived from a more fractionated magma. A positive Eu anomaly indicates that the source of ore-forming fluid is complex. The $\delta Eu$ and $\delta Ce$ differences between the Manas and East Sayan samples indicate that they underwent different geological processes and different geochemical environments in the late mineralization period.

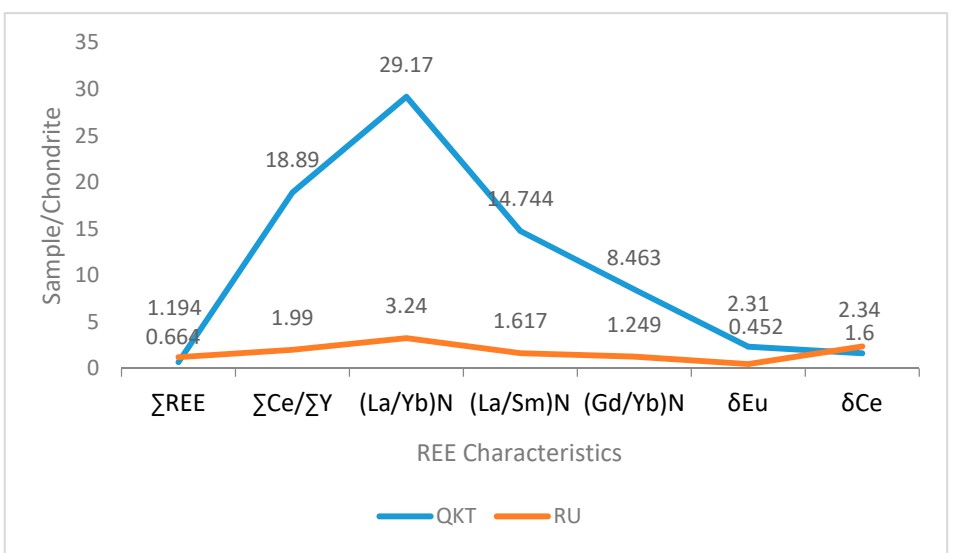

**Figure 11.** Comparison of REE parameters for the Manas and East Sayan nephrites.

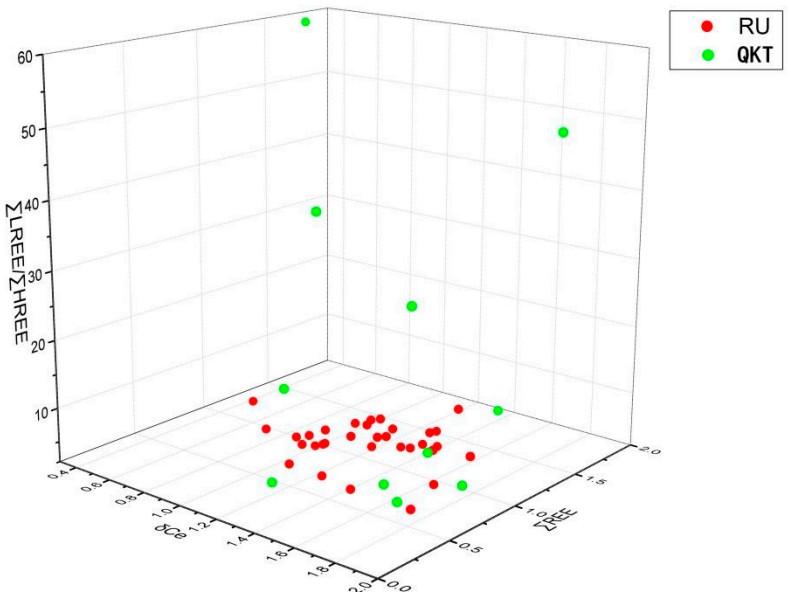

**Figure 12.** 3-D $\delta Ce$-$\sum REE$-$\sum LREE/\sum HREE$ cartesian coordinate plot for the Manas and East Sayan green nephrites.

## 4. Discussion

### 4.1. Comparison between the Russian (East Sayan) and Chinese (Manas) Green Nephrites

According to previous electron probe analyses, the main minerals of East Sayan green nephrite are tremolite and actinolite, similar to the Manas green nephrites [43]. The majority of our samples from Manas are predominantly tremolite, indicating that this is not just due to sampling bias. Chlorite-group minerals, garnet and chromite were found as impurity minerals in the Manas samples, which were also found in previous studies [44]. Chlorite-group minerals, chromite and bornite were found in the East Sayan samples, among which chlorite-group minerals and chromite (but not bornite) were found in previous studies [13,19], regardless of whether they originated from East Sayan or other green nephrite mines worldwide. Whether the bornite is a unique characteristic of the East Sayan green nephrite requires further studies.

No trace element data of China (Manas) green nephrite have been published before, and the trace element data of green nephrite from other regions in China are not comparable with this study. Russia (East Sayan) green nephrite trace element data were compiled [45] and compared (Cr, Zr, Y, Ba, Sr contents) with our data. The Cr, Zr, Y, Ba content ranges from previous studies and from this study are similar. but the content range of Manas green nephrite is different from the published data.

### 4.2. One-Way ANOVA

Various trace element contents and parameters were selected for the one-way ANOVA calculation, including: Cr, Zn, Y, Ba, Sr, δEu, $(La/Yb)_N$, $(La/Sm)_N$, $(Gd/Yb)_N$, $\sum REE$, $\sum LREE$, $\sum HREE$, $\sum LREE/\sum HREE$ and Eu/Sm. The calculation was performed with the mean, variance, 95% confidence interval of each group, sum of squares (inter-/intra-group), degree of freedom (DOF) and mean square. The values of the F statistic and significance level were obtained, which determine whether the inter-group difference is significant. An example of the calculation is shown in Table 6, and the calculation results for 14 of these factors are listed in Tables 6 and 7.

The F value is the ratio of inter- and intra-group mean squares, and increases with the significance of intergroup difference. The significance level of <0.05 is commonly used to differentiate as the standard to judge whether the differences are significant. The 95% confidence intervals of Cr, Zn, Y, Ba, Sr, δEu, $(La/Yb)_N$, $(La/Sm)_N$, $\sum HREE$, $\sum LREE/\sum HREE$ and Eu/Sm of the Manas and East Sayan samples do not overlap, and the significance level is 0. In other words, there are significant differences in the contents of Cr, Zn, Y, Ba, Sr, δEu, $(La/Yb)_N$, $(La/Sm)_N$, $\sum HREE$, $\sum LREE/\sum HREE$ and the Eu/Sm values between the Manas and East Sayan samples, which can be used to distinguish the origins of the nephrites.

**Table 6.** An example calculation of one-way ANOVA for the Cr content.

| | | | | | Mean (95% Confidence Interval) | | | |
|---|---|---|---|---|---|---|---|---|
| | | N | Mean | Std Deviation | Std Error | Lower Limit | Upper Limit | Min | Max |
| RU | | 34 | 616.2568 | 394.30407 | 67.62259 | 478.6776 | 753.836 | 11.94 | 1797.49 |
| QKT | | 23 | 81.9078 | 75.72841 | 15.79046 | 49.1604 | 114.6552 | 8.93 | 321.61 |
| Total | | 57 | 400.6423 | 404.7512 | 53.61061 | 293.2473 | 508.0372 | 8.93 | 1797.49 |
| | | | | ANOVA | | | | | |
| | | | | Quadratic Sum | DOF | Mean Square | F | Significance | |
| Between groups | (Combination) | | | 3,917,254.603 | 1 | 3,917,254.603 | 40.984 | 0 | |
| | Linear term | Unweighted | | 3,917,254.603 | 1 | 3,917,254.603 | 40.984 | 0 | |
| | | Weighted | | 3,917,254.603 | 1 | 3,917,254.603 | 40.984 | 0 | |
| Within group | | | | 5,256,863.484 | 55 | 95,579.336 | | | |
| Total | | | | 9,174,118.087 | 56 | | | | |

**Table 7.** Results of one-way ANOVA for each factor.

| Factor | Origin | 95% Confidence Interval | F | Significance |
|---|---|---|---|---|
| Cr | QKT | 49.160–114.655 | 40.984 | 0 |
| | RU | 478.678–753.836 | | |
| Zn | QKT | 28.544–33.719 | 58.026 | 0 |
| | RU | 88.397–120.286 | | |
| Y | QKT | 0.022–0.042 | 80.638 | 0 |
| | RU | 0.271–0.379 | | |
| Ba | QKT | 0.484–0.738 | 41.322 | 0 |
| | RU | 3.344–5.250 | | |
| Sr | QKT | 19.856–25.643 | 201.474 | 0 |
| | RU | 5.105–6.329 | | |
| δEu | QKT | 1.376–3.240 | 36.693 | 0 |
| | RU | 0.404–0.777 | | |
| $(La/Yb)_N$ | QKT | 13.424–44.910 | 42.794 | 0 |
| | RU | 2.567–3.912 | | |
| $(La/Sm)_N$ | QKT | 6.359–23.129 | 39.309 | 0 |
| | RU | 1.438–1.780 | | |
| $(Gd/Yb)_N$ | QKT | 0.662–16.265 | 13.654 | 0.001 |
| | RU | 1.026–1.473 | | |
| ∑REE | QKT | 0.322–1.006 | 1.24 | 0.272 |
| | RU | 0.656–1.732 | | |
| ∑LREE | QKT | 0.295–0.963 | 0.764 | 0.387 |
| | RU | 0.509–1.565 | | |
| ∑HREE | QKT | 0.014–0.056 | 43.004 | 0 |
| | RU | 0.137–0.177 | | |
| ∑LREE/∑HREE | QKT | 10.722–36.818 | 23.824 | 0 |
| | RU | 4.826–8.092 | | |
| Eu/Sm | QKT | 0.994–2.111 | 86.66 | 0 |
| | RU | 0.084–0.217 | | |

Although the 95% confidence interval of $(Gd/Yb)_N$ for the Manas and East Sayan samples overlap, the significance level is 0.001 (<<0.05). The 95% confidence intervals of ∑REE and ∑LREE for the Manas and East Sayan samples overlap, and the F values are 0.272 and 0.387 (both >0.05), respectively, suggesting that these two parameters cannot be reliably used to determine the origin of the nephrites.

## 5. Conclusions

This study compares the green nephrites from China (Manas) and Russia (East Sayan) addressing aspects of appearance, mineral component and composition, and trace elements, and presents the following findings:

1.  From their appearance, the Manas green nephrite has a low brightness, medium saturation, and bluish or grayish green color; East Sayan green nephrite has a high brightness and saturation, and green color. Some samples have a high brightness, low saturation, and greenish gray (duck-egg cyan) color. The Manas green nephrite has a sugary texture, green spots, and is coarse-grained, whereas the East Sayan green nephrite shows no sugary texture or green spots, with the texture being more delicate.
2.  Both the Manas and East Sayan green nephrites comprise mainly tremolite (with minor actinolite). The Manas green nephrites contain inclusions of chromite, chlorite-group minerals and uvarovite, whilst the East Sayan green nephrite contains inclusions of chromite, chlorite-group minerals and bornite. We suggest that bornite is uniquely found in the East Sayan green nephrites.
3.  LA-ICP-MS trace element data show that amphiboles in the Manas green nephrite contain significantly higher Sr but significantly lower Cr, Zn, Y and Ba contents than their East Sayan counterparts. The $(La/Yb)_N$, $(La/Sm)_N$ and $(Gd/Yb)_N$ values of the Manas nephrite are also higher, indicating that its ore-forming fluid/magma was more

fractionated than that of the East Sayan samples. The 3-D $\delta$Ce-$\sum$REE-$\sum$LREE/$\sum$HREE cartesian coordinate plot can be used to differentiate the green nephrites from these two origins.

4.  A one-way ANOVA calculation suggests that the contents of Cr, Zn, Y, Ba, Sr, $\delta$Eu, Eu/Sm, $(La/Yb)_N$, $(La/Sm)_N$, $(Gd/Yb)_N$, $\sum$HREE and $\sum$LREE/$\sum$HREE (but not $\sum$REE and $\sum$LREE) in the amphibole can be used to differentiate the Manas green nephrites from the East Sayan nephrites.

**Author Contributions:** Conceptualization, J.W. and G.S.; methodology, J.W. and G.S.; validation, J.W. and G.S.; formal analysis, J.W. and G.S.; investigation, J.W. and G.S.; resources, J.W. and G.S.; writing—original draft preparation, J.W.; writing—review and editing, J.W. and G.S.; supervision, G.S.; project administration, G.S.; funding acquisition, G.S. All authors have read and agreed to the published version of the manuscript.

**Funding:** This study was supported by "National Natural Science Foundation of China, grant No. 41688103,41773044" and "Second Tibetan Plateau Scientific Expedition and Research Program (STHP), grant No. 2019QZKK0802".

**Acknowledgments:** We appreciate Xuemei Zhang, Jianhong Ren, and Kong Gao for their helpful discussions and suggestions. We also thank Xinling Li, Chu Long, Jinhong Zhang, Yanhai Gao and Qiang Liu for their support of fieldwork. We are grateful to the Editors and anonymous Reviewers for their constructive and helpful comments, which significantly improved the manuscript.

**Conflicts of Interest:** The authors declare no conflict of interest.

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
