# Peer review of "Comparative Study on the Origin and Characteristics of Chinese (Manas) and Russian (East Sayan) Green Nephrites"

_minerals, doi:10.3390/min11121434_

Round 1
Reviewer 1 Report
The article presents new and interesting data on comparison of some Russians and Chinese nephrites. It is well written and properly designed.
I see three main issues, which should be solved before paper accepted.
1) Comparison with previously published data on Russians and Chinese nephrites are needed for discussion.
For example, geochemical and mineralogical features of nephrites of East Siberia were studied by Burtseva et al. 2015 (DOI 10.1016/j.rgg.2015.02.003). Chinese nephrites are extensively studied in a number of articles (e.g., 10.1016/j.oregeorev.2020.103362; 10.1016/j.oregeorev.2019.103171; 10.1016/j.oregeorev.2019.02.016)
2) ICP-MS data seems to be poorly presented, apparently contains a number of errors. Pr, Ho, Gd anomalies are clearly seen on figure 8, numbers on figure 9 are merged together, Table 4 has a wrong name, etc. I suggest to check and revise this section
3) Nephrites colours are measured by naked eye, even thought it is one of the main character of studied samples. No colour standards (e.g., Munsell colour system) were used.
Minor points:
1) Line 48. Proper name is Gorlykgol or Gorlykgolskoe (Горлыкгольское) deposit. For more names and geological background see DOI 10.1016/j.rgg.2015.02.003, DOI 10.3390/min10050394
2) Figure 2. No subsamples names are provided. No explanation for 20 subfigures for Manas and 18 subfigures for East Sayan is given.
3) Please use proper name for nephrite localities instead of "East Sayan" or Russia. There is a number of nephrite deposits in East Sayan mountains and a bigger number in Russia. Even mine number may have a sense.
4) Figure 3. It is not clear what samples are shown (as no number are present both on Fig. 2 and 3) and how they were selected. I can see that both localities have a range of colours.
5) Line 90. Please provide colour names using some standard colour scheme.
6) Line 125-126. Where these numbers are came from? Tremolite is Fe-free, but if you would like to provide average chemical composition of actionolite you need iron.
Please provide average composition of nephrits from different localities.
7) Table 1
7.1) It contains data on tremolies only, no reason to add these data in the end of the table.
7.2) Wt.% instead of Wв (check here and further in the text and in the figures)
7.3) Sample No are given randomly. It is mine No 07, 10, 02 and 10 once again. Is it possible to order them somehow? (check here and further in the Table 2)
7.4) Numebr in crystal chemical formula have 3 digits, whereas chemical analysis show only 2. Please, delete third decimal place (check here and further in the text and in the figures)
7.5) Fe3+ calculation procedure should be removed from the footnote and added to the methods. Moreover, it is not clear what is "Fe3+ maximum estimation: 13eCNK method; Fe3+ minimum estimation: 15eNK method". Please, explain. (check here and further in the Table 2)
7.6) Crystal chemical formula calculation procedure should be removed from the footnote and added to the methods. It is not clear what method was used exactly as "Calculation method: 23 oxygen atoms and 16 cations were used as reference for EPMA data, and (O+OH+F+Cl) = 24 was used as reference for chemical analysis. " (Line 129)
7.7) Sum W position=0. No OH values provided. F and Cl values provided even thought samples contain no F / Cl. (check here and further in the Table 2)
8) Figure 6 is sloppy and contain no marks to distinguish a, b, с subfigures. Beside, no explanation for "TFeO" is given.
9) Line 169. There is no such a mineral as "chlorite". You should it name it properly (e.g., clinochlore) or use chlorite group minerals.
10) Line 170. Uvarovite is missed.
11) Table 4 name seems to be erroneous (ICPS data, ppm?)
12) Figure 7 is sloppy and contain no marks to distinguish a, b, с subfigures.
13) Figure 8 show strange Pr, Ho, Gd anomalies. Please, explain. Besides, letters are very small and unreadable.
14) Figure 9. Some numbers are merged together.
15) Lines 231-238. This paragraph should be in Methods, not Discussion.
16) Line 276. Please add more data about bornite in Results and Discussion. How many grains were found? No photos of inclusion provided (SEM, BSE pictures).
Author Response
Thank you very much for your comments. Please see my responses in attachment.

Reviewer 2 Report
I suggest to the authors to reorganize the conclusions highlighting the importance of the obtained results.
I send you some specific comments and I sincerely hope that my suggestions will be useful and will help the authors to improve the manuscript.
Best Regards
Suggestions to the authors
Line 62: Please, insert a point after East Sayan;
Line 69: Please, specify “Arahushun pebble mine” for AR-p mine;
Line 85: Please change the acronym LA-HR-ICP-MS with LA-ICP-MS, which is the acronym you have used generally in the manuscript
Line 86: Please, specify better the analytical protocol of LA-ICP-MS (spot diameter? standardization? How do you recalculate the spectra? Have you used a control standard? Have you measured the precision and accuracy?) or cite the bibliography where the analytical condition normally used in the Chinese Academy of Geological Sciences are specified;
Paragraph 3.1: colorimetric analyses should be included;
Line 125: there is a mistake in the general formula of tremolite-actinolite;
Table 1: Please delete the lines F -Cl - ∑W. All numbers are 0, there is no sense to introduce these three lines;
Line 129: In Note 1, the authors specify that: FeO=FeO+Fe2O3. In this case it will be better to use “FeOtot= FeO+Fe2O3”;
Obviously, I hope that the authors have recalculated Fe2O3 as FeO before the addition!
Diagrams of figures 6 and 7: Please, delete the heading;
Line 168: Please, specify the garnet type as reported in Table 3;
Line 171: In paragraph 3.3, please, use the same acronym for the analytical technique: do you want to use the LA.HR-ICP-MS acronym or the LA-ICP-MS acronym?
Line 171: the LA-ICP-MS is a punctual analysis. Therefore it is necessary to specify where the analyses were executed (only on the tremolite crystals?)
If it is possible, it should be better to show (for example by micro-photos) the analytical points on the sample
Line 173-178 and Table 4: Please, introduce also the standard deviation and the number of analyses for each sample. When the results are higher than 100 please delete the decimals. The error is surely higher than the decimals;
Table 4: In my opinion, the caption is incorrect. Please, specify that these are the LA-ICP-MS data expressed in ppm and delete the major elements measured with EPMA from the Table (there is no sense to indicate Na, Mg, Al, Si and Ca). In any case, it is necessary to indicate the detection limit because some concentration of trace elements is very low (see for example the some values of Zr = 0.0094; Y=0.0029; Nb=0.00117 and so on).
Conclusions must be rephrased in a discursive manner.
Author Response

(The authors gave the same response as above.)

Reviewer 3 Report
The purpose of this work is to investigate the physical and chemical differences between two types of nephrites, of Russian and Chinese origin, mainly present in the Chinese market. The work is certainly interesting and worthy of publication, but it needs minor / moderate modifications.
The introduction should include a brief description of the characteristics of nephrites in general, their relation with Jades and the reasons why their study is important.
It would be necessary to describe for what kind of jewelry, or other types of materials, nephrites from Russia and China are used.
More and more in-depth information must be provided on the characteristics of the geology of the Manas and East Sayan mines.
In the paragraph on Materials and Methods the samples are indicated with the acronyms QKT and RU but is not clear what they mean. Their meaning must be reported.. Also some mines are reported with different numbers (7, 37 ..) but I don't understand what these numbers correspond to. Are they different mines in the same locality?
In the paragraph on methods it is written that microscopic observations were also conducted, but I do not see photos in the text.
The paragraph on inclusions is very short. At first did you identify the presence of inclusions through microscope? Also you could write something more about the characteristics of chlorites, if they are iron or Mg rich. Also how could you identify garnet as a uvarovite? On the base of the presence of Ca and Cr?
In the conclusions I would also report that both the Manas and East Sayan samples can be classified as tremolite (with minor actinolite) as deduced by major element analysis.
I'm not a native speaker, but I think the English form can be improved considerably. I therefore invite the authors to look at the corrections I propose in the attached PDF file

Author Response

(The authors gave the same response as above.)

Reviewer 4 Report
The name „Nephrite“ is not an offically name accredited by the IMA for a mineral!
Als Nephrit bezeichnet man einen Mischkristall aus der lückenlosen Mischreihe der Minerale Tremolit und Aktinolith. Es ist von der International Mineralogical Association (IMA) nicht als eigenständiges Mineral anerkannt.
The differences in materials investigated are possibel due to different "nephrites", as it is no mineral but a rock, the conclusions are only relevant from the material under investigation.So, the material has to be material
defintely described an identified for a Journal named "Minerals"
Author Response
The name „Nephrite“ is not an offically name accredited by the IMA for a mineral!
Als Nephrit bezeichnet man einen Mischkristall aus der lückenlosen Mischreihe der Minerale Tremolit und Aktinolith. Es ist von der International Mineralogical Association (IMA) nicht als eigenständiges Mineral anerkannt.
The differences in materials investigated are possibel due to different "nephrites", as it is no mineral but a rock, the conclusions are only relevant from the material under investigation.So, the material has to be material
defintely described an identified for a Journal named "Minerals"
Response: You are right, “Nephrite” is not a mineral accredited by IMA now. But the name “nephrite” has been used throughout. The nephrite refers to a tough, tremolite-actinolite rock with a felted, microcrystalline habit used for ornamental carvings or gems. I added an explanation in the introduction and cited a reference.

Round 2
Reviewer 2 Report
the paper can be accepted in the present form
Reviewer 4 Report
The manuscript gives data on a multi-mineral assamblage of different deposits and does not really focus on mineralogical objectives but more or less on different material characteristics which may be relevant for a certain market. Though there are lots of data and analytical research points it is more appropiate for peridicals such as "Mineralogical Record", "Gems and Gemology", Gemmologie" or "Lapis".